# A Descriptive Cohort of Suicidal Cancer Patients: Analysis of the Autopsy Case Series from 1993 to 2019 in Milan (Italy)

**DOI:** 10.3390/ijerph19020829

**Published:** 2022-01-12

**Authors:** Guendalina Gentile, Stefano Tambuzzi, Raffaella Calati, Riccardo Zoja

**Affiliations:** 1Department of Biomedical Sciences for Health, Section of Legal Medicine and Insurance, University of Milan, 20133 Milan, Italy; guendalina.gentile@unimi.it (G.G.); stefano.tambuzzi@unimi.it (S.T.); riccardo.zoia@unimi.it (R.Z.); 2Department of Psychology, University of Milan-Bicocca, 20126 Milan, Italy; 3Department of Adult Psychiatry, Nimes University Hospital, 30029 Nimes, France

**Keywords:** suicide, cancer, depression, oncology, mental health, autopsy

## Abstract

Suicide in cancer patients has always been a subject of clinical studies, but the contribution of forensic pathology to this phenomenon is poorly reported. With the aim of at least partially filling this gap in information, at the Institute of Forensic Medicine of Milan, Italy, we assessed all suicides that occurred in cancer patients. A descriptive and retrospective analysis was carried out by examining the database of the Institute and autopsy reports. We included 288 suicide cases with proven cancer diseases. For each suicide, sex, age, country of origin, body area affected by cancer, further pathological history, medications, previous suicide attempts and suicidal communications, as well as the place where the suicide occurred, were assessed. Furthermore, from a forensic point of view, we considered the chosen suicide method and any involved means. The majority of cases were male older adults affected by lung, colon and prostate cancer. Violent suicide methods were prevalent, and the most represented suicide method was falling from height regardless of the body area affected by cancer. Such data may be of clinical use for clinicians engaged in the front lines in order to address suicide risk prevention strategies among cancer patients.

## 1. Introduction

The World Health Organization (WHO) has estimated that 700,000 people die by suicide every year in the world [1] and that cancer globally causes approximately 10 million decedents [2]. A malignant tumor diagnosis can also be a stressor that may increase the likelihood of suicide [3,4,5]. In cancer patients, the risk of suicide is up to double compared to the one of the general population in the United States, Europe, Australia, Taiwan and South Korea [6,7]. Moreover, the risk is at its highest during the first months and, in any case, in the first year after diagnosis [8] or, at most, within two years [9]. An important role is played by the anatomical localization of the neoplasm [5,10,11]; indeed, tumors located in the head and neck [12] and those arising in the digestive tract (especially in the esophagus, pancreas and stomach) [13], as well as those affecting lungs [14] and prostate [15], are associated with higher rates of suicide [16]. The incidence increases in older adults [17], as well as in subjects in the progression or terminal phase of cancer [18]. The increase in suicide risk is more pronounced for men [19,20] than for women [21], especially when older than 54 years old [6], suffering from depression [22] and in the case of malignant neoplastic disease with poor prognosis [23]. Women, on the other hand, when affected by neoplasms localized in gynecological areas, show a higher incidence of suicide than those affected by tumors localized in other anatomical sites [24]. 

Although the overall cancer survival rates thanks to the progress of screening and treatment have improved [25] in Italy [26] and globally [5], cancer diagnosis [27], due to the emotional shock and upheaval that it causes or the progression of the disease [28], can represent a risk factor for suicide [6]. Indeed, numerous population studies around the world [21] have already established, over the last years [29], an increase in suicide rates in cancer patients [30], especially if depressed [22,31]. Consequently, the phenomenon of suicide in cancer patients is the subject of many studies, but the contribution of forensic pathology is poorly represented so far [3,14,32,33,34,35,36]. In particular, suicide methods of cancer patients and the site of the neoplasm have not yet been explored. Such information, however, being able to be obtained exclusively by using an autopsy approach, would inevitably be compromised or lost without an autopsy. Therefore, the privileged position from which a forensic pathologist works allows exploring, in depth, not only the causes of death but also the methods of suicide in cancer patients.

With the aim of filling at least part of this gap, we conducted a descriptive analysis of the characteristics of suicides occurring in subjects with malignant neoplasia whose autopsies were performed at the Institute of Forensic Medicine in Milan, Italy, between 1993 and 2019. The data that emerged from this cross-section of reality can both help forensic pathologists to address the often-challenging cases of suicide in cancer patients and be useful to clinicians engaged on the front lines to address suicide risk prevention strategies among cancer patients.

## 2. Materials and Methods

A descriptive and retrospective analysis was carried out on all 25,512 autopsies performed between 1993 and 2019 by using the database of the Institute of Forensic Medicine in Milan and autopsy reports (a copy of which is stored in the archives). To enroll the cases of our interest, we considered the following inclusion criteria: (i) subjects that underwent an autopsy; (ii) subjects with a proven malignant oncological pathology, based on the information available from clinical records seized by the Judicial Authority and still in our possession, as well as written in the farewell notes left behind by the decedents or reported by their family members during body identification; (iii) subjects that were aware of being affected by a cancer. With respect to the cause of death of each subject, we referred to the death certificate completed by the forensic pathologist at the end of the autopsy. We excluded all decedents that were affected from dementia, always based on the clinical records, or for whom the cancer diagnosis was not certain, as well as all cases in which a neoplastic lesion was detected for the first time during the autopsy. For each enrolled subject, we extracted the following information: sex, age, country of origin, location of cancer, further pathological history, medications, previous suicide attempts, and suicidal communications (threats and plans), as well as the location of the suicide. All data collected were analyzed using descriptive statistics. 

Furthermore, from a forensic point of view, we considered includible as suicides all cases in which circumstantial information (suicidal communications and farewell notes), police reports and autopsy evidence were consistent and converging towards a deliberately self-induced death. For all enrolled cases, we carefully assessed the suicide methods chosen by the victims and the involved means, if present. According to the methods used, suicide was classified as violent (hanging, stabbing, shooting, jumping from buildings or in front of vehicles, severe deliberate car accident, electricity and fire) or non-violent (illicit or prescription drug overdoses, CO-intoxication, plastic bag suffocation and drowning) [37,38]. Furthermore, based on the number of detrimental modalities involved in the suicide, we distinguished simple suicides (only one modality) from complex and complicated suicides. In the forensic literature, complex suicides refer to highly uncommon suicides in which multiple detrimental methods are used simultaneously or in chronological succession to avoid possible ineffectiveness of one of the chosen methods and achieving a guaranteed fatal outcome [39,40,41]. On the other hand, the term “complicated suicide” refers to a suicide in which the method established by the victim fails, but death still occurs due to an accidental secondary trauma that was not planned [42,43]. As for complex and complicated suicides, the distinction between violent and non-violent suicides is based on the first detrimental modality that was chosen and applied. During all autopsies, biological fluids (heart and femoral blood, urine, bile and gastric contents) and organ fragments (brain, lung, liver and kidney) were sampled from bodies for possible toxicological analyzes. In Italy, the analysis of cadaveric samples can be carried out only with the expressed authorization of the investigating magistrate for each individual case. In this study, all enrolled cases for which toxicological analyses were authorized were performed at the Forensic Toxicology laboratory of the University of Milan in order to detect ethyl alcohol, drugs and any other substance with pharmacological activity.

## 3. Results

### 3.1. Socio-Demographic Features

In the analyzed period of time, 4,498 suicides occurred (17.6% of all the autopsies) and, among them, 288 were subjects with diagnosed malignant neoplasm (6.40% of all suicide cases and 1.2% of total autopsies). 

Males were involved globally in 200 cases (69.4% of the total), with a sex ratio of 2.3:1.

For a description of the age distribution of the sample according to sex, refer to Figure 1. For males, the maximum and minimum age range was 28–94 years old, with mean of 69.27 and median of 73; for females, the maximum and minimum age range was 25–93 years old, with mean of 63.97 and median of 66. No victims were recorded under the age of 24. The youngest victim was a 25-year-old girl. The majority of the sample was represented by older adults: 32% of cases were 71–80-year-old (93 cases), 22% were 61–70 year old (65 cases) and 17% were 81–90 year old (49 cases). The older case was a 94-year-old man. 

Cases were mainly of Italian origins (283, 98.3%), three were white Europeans (Spanish, Albanian, Norwegian), one was Asiatic (Malaysian) and one was black African (Somalian). All such five decedents had been living in Italy for at least 5 years.

### 3.2. Cancer-Related Features 

For a detailed description of prevalent neoplastic sites according also to sex, refer to Table 1. Malignant neoplasms observed in the enrolled victims involved the following body areas: head (*n* = 14, 4.9%), with the prevalence of brain cancer; neck (*n* = 23, 8%), with the prevalence of laryngeal cancer; chest (*n* = 73, 25.3%), with the prevalence of lung cancer; abdomen (*n* = 78, 27.1%), with the prevalence of colon cancer; reproductive system (*n* = 56, 19.4%), with the prevalence of prostate cancer; urinary system (*n* = 22, 7.7%), with the prevalence of bladder cancer; immune system (*n* = 16, 5.6%); integumentary system (*n* = 5, 1.7%), skeletal system (1, 0.3%). In males, main neoplastic sites involved lung and prostate, colon, larynx and bladder; on the other hand, in females, breast, uterus and colon were the main neoplastic sites.

With respect to the correspondence between the site of the neoplasm and the victims’ decade of age, it was observed that half of the younger decedents (25–30-years-olds, mean age of 28 years old) had testicular cancer. Furthermore, lung cancer prevailed among 31–40-year-old decedents (27.3%, mean age of 36 years old); breast cancer and immune system cancers prevailed among 41–50-year-old decedents (respectively, 20%, mean age of 47.4 years old, and 16%, mean age of 44.6 years old); breast cancer and lung cancer prevailed among 51–60-year-old decedents (respectively, 21%, mean age of 54.6 years old, and 13%, mean age of 59 years old), as well as among 61–70-year-old ones (respectively, in 15.4%, mean age of 65.9 years old, and 13.4%, mean age of 64.9 years old). Prostate cancer and lung cancer prevailed among 71–80-year-old decedents (respectively, 18.3%, mean age of 76.5 years old, and 16.1%, mean age of 75.4 years old); finally, among 81–90- year-old decedents, prostate cancer (28.6%, mean age of 85.2 years old) and colon and bladder cancer (12.3% each, respectively, with mean age of 85.2 years old and 84.5 years old) prevailed.

Furthermore, in 17 cases (6% out of the total), the existence of distant metastasis was reported, starting from a primary tumor localized to lung, prostate and colon (three cases—17.6% for each); larynx and liver (two cases—11.8% for each); and testicle, tongue, skin and bladder (one case—5.9% for each).

### 3.3. Clinical Features

In addition to malignant neoplasm, 185 victims (64%) had a further positive medical history consisting of a single disease in 139 cases (75%) and multiple diseases in 46 cases (25%), the details of which are shown in Table 2.

It is clear that, in the group of a single disease, psychiatric disorders were the most prevalent, followed by organic diseases, such as cardiovascular pathologies, diabetes and chronic infections (HCV). Moreover, in the group of multiple diseases, psychiatric disorders resulted to be the most represented, since they affected the vast majority of subjects; They were associated with chronic cardiovascular, respiratory, renal and nervous diseases. Other victims did not suffer from psychiatric disorders but a combination of organic diseases, such as respiratory diseases and chronic cardiovascular or neurological pathologies. Overall, the subjects affected by depression were 146 (51% of the total).

Out of a total of all victims with a positive medical history, 201 (70%) were undergoing pharmacological treatment. On the whole, 89 subjects (44.2%) were taking drugs for psychiatric pathologies, and, among them, 67 decedents (75.2%) were taking therapy for depression; 68 (33.8%) were undergoing specific pharmacological treatment for cancer with chemotherapy. Furthermore, 133 (66%) subjects were taking only one medication, and 68 (34%) were taking multi-drug therapy. In the first group, chemotherapy was clearly the most frequently observed medication (44 cases—33%) followed by antidepressants (38 cases—28.6%); the detail of the remaining drugs taken as monotherapy is reported in Table 3. In the second group, the most frequently reported medications were antidepressants (29 cases—43%), chemotherapy (18 cases—26.5%), analgesics (8 cases—12%), diuretics (6 cases—8.8%), anxiolytics (5 cases—7.4%) and antihypertensives (2 cases—3%); the details of the multi-drug medications are reported in Table 3. As far as it is known, 68 subjects were, therefore, under chemotherapy treatment at the time of suicide. However, for subjects off therapy, the time elapsed from the off-therapy period to suicide was not available.

### 3.4. Suicide-Related Features 

Of the 288 enrolled cases, 119 subjects (41.3%) had previously communicated that they wanted to die by suicide, 48 subjects (16.5%) had made previous suicide attempts and 42 (14.6%) had adopted both behaviors. The main reasons reported in the farewell notes or by the relatives for the suicide were the poor efficacy of the medical treatments (*n* = 134), serious side effects of therapies (*n* = 236), the worsening of the quality of life (*n* = 263), derived surgical complications (*n* = 57), insufficient pain management (*n* = 123), the progression of the disease (*n* = 191), poor prognosis (*n* = 245) and, ultimately, related physical deteriorations (*n* = 251). As expected, there were often multiple coexistent reasons behind the decision to die from suicide.

The majority of suicide cases (*n* = 283, 98.3%) used a single suicide method and the minority used two different methods simultaneously or in a chronological sequence (*n* = 4, 1.4%), carrying out a complex suicide. In detail, complex suicides involved only male subjects and consisted of the following: hanging and plastic bag suffocation; hanging and gunshot wound; sharp injuries and plastic bag suffocation; and sharp injuries and falling from a height, in one case each. A complicated suicide occurred in only one case (0.3%), which was accomplished by failed hanging followed by a fall from height.

A violent suicide was present in 251 cases (246 simple, 4 complex and 1 complicated suicide) for a total of 87%, while a non-violent suicide was recorded in 37 cases (all simple suicides) for a total of 13% (Table 4). In detail, 181 males (72%) and 70 females (28%) were involved in violent suicide; on the other hand, 19 males (51.5%) and 18 females (48.5%) were involved in non-violent suicide.

To better understand the phenomenon of suicide in cancer patients by using forensic data, we have briefly related prevalent neoplastic sites to detrimental modalities applied to death by suicide, maintaining the distinction between males and females (Table 5). 

With respect to toxicological analysis, they were authorized by the investigating magistrates in 169 cases. In 60 subjects (35.5%), the blood alcohol level (BAC) was found between 0.1 and 0.8 g/L. The dosage of carboxyhemoglobin (HbCO) was required for all six victims who died of suicide through carbon monoxide inhalation, and in all such cases the concentrations of HbCO higher than 33% were measured in their heart blood. In the literature, this value is considered as a threshold for acute and potentially lethal CO intoxication [44]. Regarding illicit drugs, only four victims tested positive for cannabinoids. Finally, traces of caffeine and cotinine were identified in 103 victims.

Overall, in all 288 cases, a cause of death was determined, also thanks to the performance of lab analyses. Thus, in no case was the cause of death undetermined, similarly to what normally occurs in all suicide cases that undergo autopsy.

Finally, we also analyzed the location where suicides took place. It was observed that 66% of such events occurred in the victims’ house, followed by 30% of cases by outdoor locations, among which the hospital courtyard and urban streets were the most prevalent; the remaining 4% of victims died by suicide in closed places outside their houses, especially in hospital rooms (Figure 2).

## 4. Discussion

We performed a descriptive and retrospective post-mortem analysis on suicides in subjects suffering from a malignant neoplasm (*n* = 288) in the database of the Institute of Forensic Medicine in Milan, Italy, over 27 years (1993–2019). In our study, in the majority of cases, victims were males and older adults, with a mean age of 66.6 years, reaching the maximum number of cases in the decade between 71 and 80 years. Decades between 25 and 41 years old were the least represented. These findings are consistent with the data reported in a recent literature review [45]. In our cases, we have not recorded suicides of pediatric cancer patients, although this phenomenon is well documented in the literature even in cancers with long-term survival [46]. However, one of the inclusion criteria of our work was that all subjects must have undergone an autopsy. In fact, the performance of autopsies on minors (children and adolescents) is a practice that may not be routinely performed or tends to be discouraged. It is too often a procedure considered “inappropriate” as it maims the innocence of younger victims according to certain cultures [47].

With respect to the specific viscera, which were the site of tumor development, lungs were the most prevalent, followed by the prostate, breast and colon, similarly to what is reported by other international studies [48]. To better understand our data, we took into consideration the prevalence of cancers in the general Italian population. We found that, in descending order, lung, breast, prostate and colon cancers are the most frequently diagnosed in the general population [49]. These data are entirely consistent with the prevalence of cancers recorded in our analysis. The only discrepancy we observed concerned a higher prevalence of prostate cancer compared to breast cancer. However, this is a completely explainable finding considering the higher number of male victims compared to the female ones in our case series (ratio of 2.3 to 1).

In addition to malignant neoplasm, the majority of the subjects (64%) were also affected by other diseases, particularly depression (51%), which is notoriously more present in cancer patients than in the general population [5]. Such findings were coherent with the data reported in the literature [32]. As for depression, it emerged that only 67 (46%) out of the total of 146 subjects were in specific pharmacological treatment. This discrepancy can only be minimally explained by a lack of clinical and health information relating to the enrolled victims. Conversely, it emphasizes the fact that most likely greater attention should be paid to screening of depressive symptoms and the treatment of this pathology, especially in a category of subjects at high risk of suicide, such as cancer patients. Furthermore, in such patients, the management of depression and suicide prevention strategies should be modulated based on the illness trajectory. Thus, the literature suggests that different prevention strategies may be developed with respect to three different periods within the disease course. The first would target patients who have recently been diagnosed with cancer. The second one would be focused on cancer patients under treatment or cancer survivors, who may suffer from functional loss due to invasive anti-cancer treatments. The last one would be for patients with advanced cancer [50,51]. Studies have consistently shown that not only major depression and emotional distress but also substantial pain, impaired physical functioning and the loss of autonomy and independence are clinical factors most commonly associated with suicidal thoughts in these patients [52,53,54,55,56]. The preponderant role of depression is supported by the fact that in cancer patients the risk of suicide tends to disappear after an adequate treatment of depression [50,51]. Therefore, it is conceivable that adequate antidepressant therapy could have prevented suicide in at least part of the cancer patients enrolled in our study. It is usually thought that only cancers with a poor prognosis have a greater impact on the quality of life. However, it is important to point out that even curable cancers, such as testicular germ cell tumors, or those with long-term survival, such as low-grade gliomas, can severely affect patients’ quality of life or self-image and cause psychiatric stress, which can result in suicide [46].

As for the method of suicide, violent suicides (87%) prevailed by far compared to non-violent ones (13%). The male gender was far more involved in violent suicides (72%). As for non-violent suicides, the two genders were almost overlapping, with a very slight male prevalence equal to 51.5% against 48.5%. Only for one detrimental modality, i.e., the chemical one, was a greater involvement of the female sex observed.

By comparing anatomical sites affected by cancer with the applied suicide method, the most observed detrimental modality was the fall from height, regardless of the involved body area. Suicidal gunshots were mostly performed by subjects affected by lung and prostate cancer. On the other hand, hanging was mostly preferred by subjects suffering from lung and colon cancer. Among non-violent suicides, the chemical modality prevailed, especially in patients affected by breast cancer. With respect to chemical injury, the suicidal intake of chemical substances, considered as a “sweet and painless death”, could have been easily applied by subjects in possession of many categories of medications (analgesics, morphine and benzodiazepines); however, it concerned only a minority of all cases enrolled in the study. In particular, we did not record any victim who died from an opioid drug overdose. This finding is in contrast to what has been reported in the literature, especially in countries such as the United States, where the phenomena of opioid dependence and abuse are mainly related to their non-medical use [57]. In Italy, the use of opioid analgesics remains far lower than in Northern Europe and the United States. In addition, numerous efforts have been made to ensure that physicians and pharmacists exercise greater vigilance in prescribing or dispensing opioid drug to at-risk patients [58]. For these reasons, most likely in cancer patients, the use of opioid drugs remains more restricted to medical use only. In general, when examining the violent or non-violent nature of the suicidal modality, it appears to be explained by its relationship to the gender of the victims rather than to the site of the tumor.

Most likely, the choice of the largest number of cancer patients to die by suicide by falling from a height was dictated by some intrinsic characteristics of such a detrimental modality, such as the simplicity of performance and the easy accessibility to windows and balconies; moreover, it does not require subject’s familiarity and practicality of any tool and, above all, it lends itself very well to an impulsive gesture. On the contrary, all other suicide methods imply different-degree programming. Such a hypothesis is also supported by the fact that almost all cancer patients who die by suicide in the hospital setting decided to resort to falling from a height. Although we are not aware of the reason why such patients were in the hospital the day they died, they had probably visited for problems related to their neoplastic pathology. Considering these results, restricting access to suicide means of places in general hospitals that may become hotspots must be considered. On the other hand, compared to the majority of cancer patients who died by suicide after an apparently impulsive gesture, a minority of them seemed to have carried out planned suicides. Among such subjects, some resorted to different detrimental modalities simultaneously or in chronological succession. The purpose was to avoid a possible ineffectiveness of one of the chosen mechanisms and achieve a guaranteed fatal outcome testifying their determination to die [39,41]. In relation to emerged data, for the sake of completeness, we emphasize that, in Italy, currently, there is still no legislation that regulates physician-assisted suicide, unlike many other European countries.

## 5. Conclusions

Although the carried-out analysis is of a descriptive nature, our data allowed outlining possible “typical” characteristics of cancer patients who died by suicide: they are mostly elderly men that are often depressed and affected by a malignant neoplasm with predominantly thoracic or abdominal localization (lung, breast, colon and prostate). They usually resort to a violent suicide method, thus being able to configure cases of challenging classification regarding the modality of death for forensic pathologists. In view of the documented link between cancer patients and violent suicide, forensic pathologists should be aware of the high chance to find even very destructive injuries in these victims without necessarily implying a method other than suicide. In the possibility of examining in detail both the cause of death and the modalities of suicide of oncological patients lies the precious and alternative contribution that forensic pathology can offer to shed further light on this phenomenon. It is, in fact, information of exclusive autopsy-forensic origin. In this sense, autopsy stands as an indispensable means of protection of public health, and prevention for the entire community. Indeed, the considerations that emerged from this descriptive analysis could certainly have important repercussions. They could serve both to raise public awareness on the phenomenon of suicide in cancer patients and help deepen and improve the clinical approach of psychological, psychiatric and oncological services in identifying adequate prevention tools and in offering patients effective cancer interventions specifically targeted at suicide risk.

### Limitations

We acknowledge that anamnestic data of the victims enrolled in this retrospective study may be incomplete in some of their parts, especially with respect to cancer patients’ degree of physical, social and emotional suffering, as well as their feelings with respect to the oncological pathology they were suffering from. Further consideration is the lack of comparison with a cohort of suicides of non-oncological subjects. This aspect is absent because we conducted descriptive analysis of the phenomenon of suicide in cancer patients from an autopsy-forensic point of view, although such a comparison could represent a prospect for future investigation. Similarly, the possibility of conducting a study of suicides in patients first diagnosed with cancer at autopsy could also be an interesting research development in order to investigate possible correlations between systemic inflammatory mechanisms, neoplasia, depression and suicide.

## Figures and Tables

**Figure 1 ijerph-19-00829-f001:**
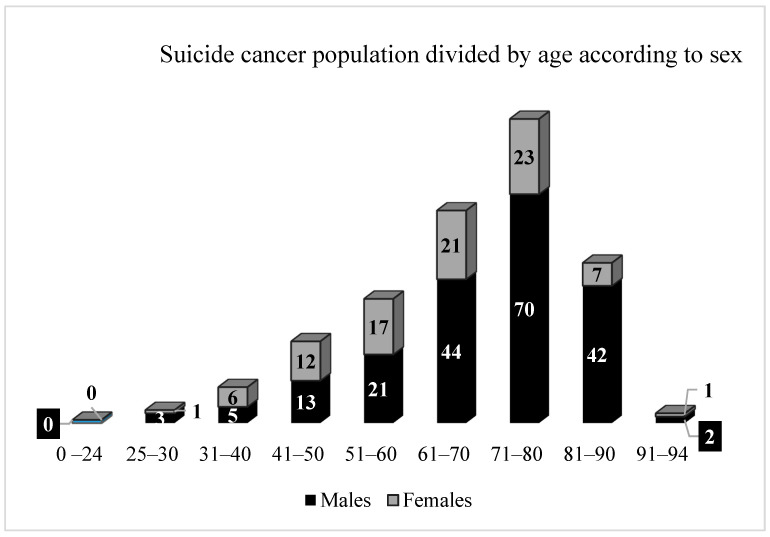
Graphic representation of the subjects died by suicide affected with cancer divided by sex and age decades.

**Figure 2 ijerph-19-00829-f002:**
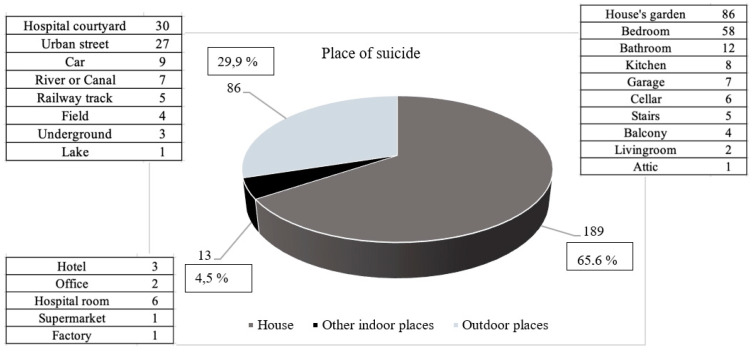
Graphic representation of locations where suicides occurred.

**Table 1 ijerph-19-00829-t001:** Details by anatomical district of the prevalent neoplastic sites according to the victim’s sex.

Body Area	Site	Total (%)	Sex	
			M (%)	F (%)
Head 14 (4.9%)	Brain	9 (64.3)	5 (55.5)	4 (44.5)
	Ear	1 (7.1)	1 (100)	
	Tongue	4 (28.6)	3 (75)	1 (25)
Neck 23 (8%)	Pharynx	2 (8.8)	2 (100)	
	Thyroid	6 (26)	2 (33.3)	4 (66.7)
	Larynx	15 (65.2)	14 (93.3)	1 (6.7)
Chest 73 (25.3%)	Esophagus	4 (5.5)	4 (100)	
	Breast	29 (39.7)		29 (100)
	Lung	40 (54.8)	36 (90)	4 (10)
Abdomen 78 (27.1%)	Liver	11 (14.1)	7 (63.6)	4 (36.4)
	Biliary tract	4 (5.1)	4 (100)	
	Pancreas	13 (16.7)	10 (77)	3 (33)
	Stomach	12 (15.3)	11 (91.6)	1 (8.4)
	Duodenum	1 (1.4)	1 (100)	
	Colon	26 (33.3)	16 (61.5)	10 (38.4)
	Rectum	9 (11.6)	6 (66.6)	3 (33.4)
	Peritoneum	2 (2.5)	2 (100)	
Reproductive system 56 (19.4%)	Testicle	5 (8.9)	5 (100)	
	Prostate	36 (64.3)	36 (100)	
	Ovary	3 (5.3)		3 (100)
	Uterus	12 (21.5)		12 (100)
Urinary system 22 (7.7%)	Kidney	10 (45.5)	7 (70)	3 (30)
	Bladder	12 (54.5)	11 (91.6)	1 (8.4)
Immune system 16 (5.5%)	Lymphocytes	16 (100)	12 (75)	4 (25)
Integumentary system 5 (1.7%)	Skin	5 (100)	5 (100)	
Skeletal system 1 (0.3%)	Bone	1 (100)	1 (100)	

**Table 2 ijerph-19-00829-t002:** Details of single and multiple disease affecting victims.

Type of Pathology		Disease 1	Disease 2	*N* of Cases	%
**Single disease—139**	Psychiatric 111 (79.8%)	Depression		109	98.1
		Psychosis		2	1.9
	Organic 28 (20.2%)	Hypertension		9	32.1
		Myocardial infarction		8	28.5
		Coronary artery disease		2	7.2
		Diabetes		7	25
		Hepatitis c virus (HCV)		2	7.2
**Multiple diseases—46**	Psychiatric 37 (80.4%)	Depression 37 (100%)	Alcoholism	8	21.7
			HCV	4	10.8
			Myocardial infarction	4	10.8
			Hypertension	4	10.8
			Diabetes	3	8.1
			Arrhythmia	3	8.1
			Bronchial asthma	2	5.4
			Kidney failure	2	5.4
			Maculopathy	2	5.4
			Epilepsy	1	2.7
			Cerebral stroke	1	2.7
			Parkinson disease	1	2.7
			Pulmonary emphysema	1	2.7
			Sarcoidosis	1	2.7
	Organic 9 (19.6)	COPD (chronic obstructive pulmonary disease) 4 (44.4%)	Hypertension	2	50
			HCV	1	25
			Heart failure	1	25
		Hypertension 5 (55.6)	Aortic aneurysm	1	20
			Diabetes	1	20
			Epilepsy	1	20
			Glaucoma	1	20
			Neuropathy	1	20

**Table 3 ijerph-19-00829-t003:** Medications taken by the enrolled victims. In A, details of single-drug therapy; in B, details of multi-drug therapy.

**(A)**			
**Medication**			***N* of Cases (%)**
**Single-drug therapy—133–66%**		Chemotherapy	44 (33)
		Antidepressants	38 (28.7)
		Anxiolytics	12 (9)
		Antihypertensive	5 (3.8)
		Analgesic	8 (6)
		Antiepileptic	2 (1.5)
		Oxygen therapy	1 (0.7)
		Antipsychotics	4 (3)
		Proton-pump inhibitor	2 (1.5)
		Antithyroid	3 (2.3)
		Antiparkinsonian	1 (0.7)
		Antiplatelets agents	3 (2.3)
		Oral hypoglycemic agents	3 (2.3)
		Anti-inflammatory	4 (3)
		Diuretics	2 (1.5)
		Blood products	1 (0.7)
**(B)**			
**Medication**			***N* of Cases (%)**
**Multi-drug therapy—68–34%**	Antidepressants 29 (42.7%)	Chemotherapy	6 (20.6)
		Anxiolytics	13 (44.8)
		Antihypertensive	4 (13.8)
		Morphine	3 (10.3)
		Immunosuppressants	1 (3.5)
		Antiarrhythmic	1 (3.5)
		Antiepileptic	1 (3.5)
	Chemotherapy 18 (26.5%)	Morphine	5 (27.7)
		Antihypertensive	4 (22.2)
		Insulin	2 (11.2)
		Cortisone	2 (11.2)
		Diuretics	1 (5.5)
		Antiemetics	2 (11.2)
		Interferon	1 (5.5)
		Oral hypoglycemic agents	1 (5.5)
	Analgesic 8 (11.8%)	Antiplatelets agents	3 (37.5)
		Cortisone	2 (25)
		Diuretics	2 (25)
		Proton-pump inhibitor	1 (12.5)
	Diuretics 6 (8.8%)	Cardiotonic	2 (33.6)
		Oral hypoglycemic agents	1 (16.6)
		Antiepileptic	1 (16.6)
		Anxiolytics	1 (16.6)
		Antiplatelets agents	1 (16.6)
	Anxiolytics 5 (7.3%)	Cortisone	1 (20)
		Sedative-hypnotic	4 (80)
	Antihypertensive 2 (2.9%)	Antiplatelets agents	1 (50)
		Cortisone	1 (50)

**Table 4 ijerph-19-00829-t004:** Details of violent and non-violent suicides.

Detrimental Modalities and Applied Means According to Victims’ Sex and Cause of Death in Simple (Nor Complex, Complicated) Suicides											
	Detrimental Modality	Applied Mean	Total	%	Sex				Cause of Death	Total	%
					M	%	F	%			
Violent suicides—246 cases 85.5%	Falling from a height (3–27 m)		145	94.8	87	60	58	40	Skeletal-visceral injuries	138	95.1
									Brain injuries	7	4.9
	Pedestrian running over	Train (5 cases—62.5%)	8	5.2	7	87.5	1	12.5	Body disruption	6	75
		Subway (3 cases—37.5%)							Brain disruption	2	25
	Gunshot	Gun (46 cases—88.5%)	52	18	48	92.3	4	7.7	Brain injuries	46	88.4
		Shotgun (6 cases—11.5%)							Cardio-pulmonary injuries	5	9.6
									Acute bleeding due to vascular lesions	1	2
	Hanging	Rope (17 cases—50%)	34	12	30	88.2	4	11.8	Hanging mechanical asphyxia	34	100
		Electric cable (6 cases—17.5%)									
		Roller shutter belt (1 case—3%)									
		Mountaineer rope (1 case—3%)									
		Clothing (7 cases—20.5%)									
		Scarf (1 case—3%)									
		Shoelaces (1 case—3%)									
	Sharp force trauma	Knife (5 cases—71.4%)	7	2.5	4	57.1	3	42.9	Acute bleeding due to vascular lesions	4	57.1
									Internal submersion	1	14.3
		Razor (1 case—14.3%)							Cardio-pericardial injuries	1	14.3
		Scalpel (1 case—14.3%)							Lung injuries	1	14.3
Non-violent suicides—37 cases 14.5%	Chemical	Antidepressants (9 cases—39%)	23	8	11	48.2	12	51.8	Acute drug poisoning	14	60.8
		Anxiolytics (3 cases—13%)									
		Insulin (2 cases—8.8%)									
		Acid caustic substances (2 cases—8.8%)							Esophagus-gastric perforation	3	13.2
		Acid caustic substances (1 case—4.4%)									
		Carbon monoxide (CO) (6 cases—26%)							Acute carbon monoxide intoxication	6	26
	Drowning	River (7 cases—87.5%)	8	3	4	50	4	50	Asphyxia by drowning	8	100
		Lake (1 case—12.5%)									
	Plastic bag suffocation	Plastic bag (6 cases—100%)	6	2	4	66.6	2	33.4	Plastic bag suffocation	6	100

**Table 5 ijerph-19-00829-t005:** Breakdown of victims based on sex, tumor site and applied detrimental suicide modality.

		Violent Suicide										Non-Violent Suicide									
Body Area	Site	Falling from a Height		Pedestrian Run over		Gunshot		Hanging		Sharp Force Trauma		Chemical		Drowning		Plastic Bag Suffocation		Complex Suicide		Complicated Suicide	
		M	F	M	F	M	F	M	F	M	F	M	F	M	F	M	F	M	F	M	F
**Head**	Brain	2	3			2		1							1						
	Ear					1															
	Tongue		1			1												1		1	
**Neck**	Pharynx					2															
	Thyroid	1	3										1		1						
	Larynx	6	1			4		2								2					
**Chest**	Esophagus	2				2															
	Breast		18				1		3				4		1		2				
	Lung	18	3	1		9		5		2	1	1									
**Abdomen**	Liver	6	2				1						1					1			
	Biliary tract	4																			
	Pancreas	6	1			2		2					2								
	Stomach	7	1			1		2						1							
	Duodenum	1																			
	Colon	7	9			1		5			1	2		1							
	Rectum	1	2	1		1	1	3													
	Peritoneum	2																			
**Reproductive System**	Testicle			1		1		1				2									
	Prostate	11		2		12		4		2		2				2		1			
	Ovary		3																		
	Uterus		9										2		1						
**Urinary System**	Kidney	3			1	1	1	1	1			1		1							
	Bladder	4	1	1		3		2				1									
**Immune System**	Lymphocytes	4	1	1		5					1	1	2					1			
**Integumentary System**	Skin	2				1		2													
**Skeletal System**	Bone	1																			

## Data Availability

All the data have been reported in the manuscript.

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
