# Peer review of "A Descriptive Cohort of Suicidal Cancer Patients: Analysis of the Autopsy Case Series from 1993 to 2019 in Milan (Italy)"

_ijerph, 2022, doi:10.3390/ijerph19020829_

Round 1

Reviewer 1 Report

Gentillet et al proposed a manuscript investigating potential features associated with suicide in cancer patients. 

The main goal was to describe characteristics aiming to define cancer patients harboring higher risk for suicide and to propose specific intervention. 

authors used a retrospective autopsy serie of 288 cases. In general, data are clearly presented, the organization is well adapted.

Some limitations could be addressed : 

  • Section 3.1 :Age : data might be presented by median or mean (range min to max and not by SD)
  • lines 137 - 146 : main age data should be compared with mean age by histology. i.e. mean age of 31-40 y.o is uncommon and this should be accurately noticed. 
  • line 147 : only 6% were cancer patients with distant metastasis and this is not attempted for a suicidal serie.. Moreover, this is not concordant with chemotherapy ongoing rate (33.8% line 167). This is disturbing and this difference should be discussed.
  • Any case of suicide by medical drug are reported, especially with no opioid (very often presribed in this context) : could the authors explain this observation and discuss it ? 
  • Table 5 is not usefull. Each line represent only one case. This should be include in text form if usefull (not sure to that the accurate description is usefull)
  • Why authors chose to restrict their analysis to patients with a past autopsy ? what is the interest 
  • Finally and as a main limitation, a comparison between this cancer cohort and with a control "suicid cohort" without cancer history might be of interest to meet reader's expectation, trying to highlight specificity for cancer patients harboring higher suicide risk. 

To summarize, the current results do not meet with the presented objective. This is currently a decriptive cohort of suicide for cancer patients with their respective modalities. I advise strongly to modify title, objective, discussoin and conclusion to meet with current reported results. 

Author Response

Reviewer 1

Gentillet et al proposed a manuscript investigating potential features associated with suicide in cancer patients. 

The main goal was to describe characteristics aiming to define cancer patients harboring higher risk for suicide and to propose specific intervention. 

authors used a retrospective autopsy serie of 288 cases. In general, data are clearly presented, the organization is well adapted.

Some limitations could be addressed : 

  • Section 3.1 :Age : data might be presented by median or mean (range min to max and not by SD)

Thank you for the comment. We followed the suggestion of the reviewer.

  • lines 137 - 146 : main age data should be compared with mean age by histology. i.e. mean age of 31-40 y.o is uncommon and this should be accurately noticed. 

Thank you for the comment. We are not sure if we understood the reviewer's request correctly, however, we have added the mean ages for each age range and tumor type. We hope we have reposted as desired.

  • line 147 : only 6% were cancer patients with distant metastasis and this is not attempted for a suicidal serie. Moreover, this is not concordant with chemotherapy ongoing rate (33.8% line 167). This is disturbing and this difference should be discussed.

We confirm that 6% of patients had distant metastases. We are sorry that we did not understand what the reviewer meant by "and this is not attempted for a suicidal series." Regarding the apparent discordance with the 33.8% proportion of patients taking chemotherapy (line 167), we point out that the data are correct. In fact, not only subjects with distant metastases took chemotherapy, as the latter was also used in case of localized tumors, for neoadjuvant as well as adjuvant purposes.

  • Any case of suicide by medical drug are reported, especially with no opioid (very often prescribed in this context) : could the authors explain this observation and discuss it ? 

Thank you for the comment. We have addressed this point in the discussion as requested.

  • Table 5 is not usefull. Each line represent only one case. This should be include in text form if usefull (not sure to that the accurate description is usefull)

Thank you for your comment. We accept the suggestion that table 5 is not useful. Therefore we have removed it from the manuscript. We have included the essential information in the text.

  • Why authors chose to restrict their analysis to patients with a past autopsy ? what is the interest 

The aim of the study we conducted was to provide a descriptive analysis of the phenomenon of suicide in cancer patients from an autopsy-forensic point of view. For this reason, the sample of enrolled individuals could not be other than subjects subjected to autopsy at our Institute of Forensic Medicine.

  • Finally and as a main limitation, a comparison between this cancer cohort and with a control "suicid cohort" without cancer history might be of interest to meet reader's expectation, trying to highlight specificity for cancer patients harboring higher suicide risk.

Thank you for the comment. We consider this observation of great relevance and agree it would be a very interesting prospect for future work. In our case, we performed only one descriptive analysis that could provide insight into the phenomenon of suicide in cancer patients from an autopsy-forensic perspective. We recognize, however, that the lack of comparison with a cohort of suicides in non-oncological individuals could be a limitation, which is why we decided to make this explicit in the limitations of the study.

To summarize, the current results do not meet with the presented objective. This is currently a decriptive cohort of suicide for cancer patients with their respective modalities. I advise strongly to modify title, objective, discussoin and conclusion to meet with current reported results.

We agree with the reviewer that our work is a descriptive analysis presentation of a cohort of suicidal cancer patients. This was our goal, approaching this phenomenon from an autopsy-forensic perspective. We therefore welcome the reviewer's suggestion to modify the title, purpose, discussion, and conclusion of the manuscript to make them appropriate for a descriptive study.

Reviewer 2 Report

This retrospective case series of autopsy results of decedents who died by suicide with a diagnosis of a malignant cancer were studied to determine the characteristics of the suicides and the relationship with the site of the neoplasm. The study results importantly highlighted the relationship between death by suicide and co-morbid depression in cancer patients; however, the following issues limited the impact of this research:
1.    The authors suggested that forensic pathologists could provide an unique perspective to understanding suicide in decedents with cancer. Readers from other disciplines may need the authors to articulate what this unique perspective can provide to understanding suicide?
2.    The study excluded cases of death by suicide where cancer lesions were detected for the first time at autopsy. This might be a very interesting group of cases to study to unravel possible mechanisms connecting neoplasms to depression to suicide; for example, the growing interest in inflammatory mechanisms as a possible link between these clinical problems. The authors could raise this issue as a limitation and area for future study.
3.    What was done about cases where the cause of death was undetermined? A certain number of undetermined cases may be suicides and often these deaths are included when studying autopsy results of death by suicide.
4.    The authors should avoid value-laden terms such as “suicide gestures” and clarify whether the term refers to expressions of suicide ideation or suicide behaviors? Also referring to “death by suicide” is preferred over “committed suicide”.
5.    The association between violent suicide deaths and site of the cancer appeared to be explained by the relationship between gender and violent suicides rather than site of the cancer.
6.    The authors should have discussed the prevalence of death by cancer site in the overall population in order to better understand the result found in decedents with cancer who died by suicide.
7.    Was physician assisted death available to Milan residents during any of the years studied? The availability of physician assisted death may impact whether patients with cancer sought this means of death versus suicide.

Author Response

Reviewer 2

This retrospective case series of autopsy results of decedents who died by suicide with a diagnosis of a malignant cancer were studied to determine the characteristics of the suicides and the relationship with the site of the neoplasm. The study results importantly highlighted the relationship between death by suicide and co-morbid depression in cancer patients; however, the following issues limited the impact of this research:
1.    The authors suggested that forensic pathologists could provide an unique perspective to understanding suicide in decedents with cancer. Readers from other disciplines may need the authors to articulate what this unique perspective can provide to understanding suicide? 

Thank you for your comment. The reviewer raised a crucial point in the paper and we are pleased to clarify this further. Therefore, we have implemented it in both the introduction and conclusion sections of the manuscript.

  1.    The study excluded cases of death by suicide where cancer lesions were detected for the first time at autopsy. This might be a very interesting group of cases to study to unravel possible mechanisms connecting neoplasms to depression to suicide; for example, the growing interest in inflammatory mechanisms as a possible link between these clinical problems. The authors could raise this issue as a limitation and area for future study.

Thank you for your comment, we agree with the reviewer that this is certainly an interesting issue and worthy of future exploration. We welcome this comment and have added this aspect within the limitations.
3.    What was done about cases where the cause of death was undetermined? A certain number of undetermined cases may be suicides and often these deaths are included when studying autopsy results of death by suicide.

Thank you for the comment. The reviewer raised an important point. We specified in the results section (suicide-related characteristics) that in none of the 288 cases was the cause of death undetermined, even thank to lab analysis. In our experience, this is true not only for the enrolled cohort of suicidal cancer patients but also for all suicides in general that undergo autopsy.

  1.    The authors should avoid value-laden terms such as “suicide gestures” and clarify whether the term refers to expressions of suicide ideation or suicide behaviors? Also referring to “death by suicide” is preferred over “committed suicide”.

Thank you for your report. We have removed the unwanted terms from the manuscript.
5.    The association between violent suicide deaths and site of the cancer appeared to be explained by the relationship between gender and violent suicides rather than site of the cancer.

Thank you for your comment. We agree with the reviewer's statement and have made this more explicit in the discussion section of the manuscript.

  1.    The authors should have discussed the prevalence of death by cancer site in the overall population in order to better understand the result found in decedents with cancer who died by suicide.

Thank you for the suggestion. As requested, to better understand our data, we have implemented the discussions by adding the prevalence of the most frequent cancers in the general population.
7.    Was physician assisted death available to Milan residents during any of the years studied? The availability of physician assisted death may impact whether patients with cancer sought this means of death versus suicide.

Thank you for the interesting comment. We would like to underline that in Italy there is still no law regulating physician-assisted suicide. Consequently, no case has been registered.

Round 2

Reviewer 1 Report

The authors have addressed the reviewer's concerns.

Reviewer 2 Report

The authors have addressed the reviewer's concerns.